# Evaluation of Cytotoxicity, Release Behavior and Phytopathogens Control by Mancozeb-Loaded Guar Gum Nanoemulsions for Sustainable Agriculture

Ravinder Kumar [1,*], Manju Nehra [2], Dharmender Kumar [3], Baljeet Singh Saharan [4], Prince Chawla [5], Pardeep Kumar Sadh [1], Anju Manuja [6] and Joginder Singh Duhan [1,*]

1 Department of Biotechnology, Chaudhary Devi Lal University, Sirsa 125055, India
2 Department of Food Science and Technology, Chaudhary Devi Lal University, Sirsa 125055, India
3 Department of Biotechnology, Deenbandhu Chhotu Ram University of Science and Technology, Murthal 131039, India; dkbiology@gmail.com
4 Department of Microbiology, CCS Haryana Agricultural University, Hisar 125004, India; baljeetsaharan@hau.ac.in
5 Department of Food Technology and Nutrition, School of Agriculture, Lovely Professional University, Jalandhar 144411, India
6 ICAR-National Research Centre on Equines, Hisar 125001, India
* Correspondence: rsulakh@gmail.com (R.K.); duhanjs68@gmail.com (J.S.D.); Tel.: +91-9416072588 (R.K.); +91-9416725009 (J.S.D.)

**Abstract:** Chemical fungicides are the backbone of modern agriculture, but an alternative formulation is necessary for sustainable crop production to address human health issues and soil/water environmental pollution. So, a green chemistry approach was used to form guar gum nanoemulsions (NEs) of 186.5–394.1 nm containing the chemical fungicide mancozeb and was characterized using various physio-chemical techniques. An 84.5% inhibition was shown by 1.5 mg/mL mancozeb-loaded NEs (GG-1.5) against *A. alternata*, comparable to commercial mancozeb (86.5 ± 0.7%). The highest mycelial inhibition was exhibited against *S. lycopersici* and *S. sclerotiorum*. In tomatoes and potatoes, NEs showed superior antifungal efficacy in pot conditions besides plant growth parameters (germination percentage, root/shoot ratio and dry biomass). About 98% of the commercial mancozeb was released in just two h, while only about 43% of mancozeb was released from nanoemulsions (0.5, 1.0 and 1.5) for the same time. The most significant results for cell viability were seen at 1.0 mg/mL concentration of treatment, where wide gaps in cell viability were observed for commercial mancozeb (21.67%) and NEs treatments (63.83–71.88%). Thus, this study may help to combat the soil and water pollution menace of harmful chemical pesticides besides protecting vegetable crops.

**Keywords:** guar gum nanoemulsions; antifungal; toxicity; slow-release; sustainable

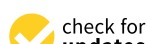



## 1. Introduction

Population explosion emerged as one of the significant challenges for feeding the growing population, particularly for the sustainable production of food and vegetables. Globally, the population may stretch up to 10.9 billion by 2100 and increase the demand for food and vegetables by nearly 50% [1]. For this increasing population, worldwide pesticide consumption will rise up to 3.5 tonnes [2]. The primary damaging fungal pathogens of tomatoes and potatoes crops are *Alternaria solani*, *Alternaria alternata*, *Stemphylium lycopersici* and *Sclerotinia sclerotiorum*. *A. solani* can cause early blight in potatoes. The disease is devastating and can cause yield losses of up to 40% [3]. Early blight signs include darkish brown to black necrosis on the leaves, followed by chlorosis. Due to the incidence of point mutations caused by the modification in *A. solani*, several *Alternaria*-specific pesticides with single-site action mechanisms developed resistance. For example, a researcher described the F129L alteration in the *cytb* gene, in decreased sensitivities to broad-spectrum fungicides

with in vitro and in vivo studies [4]. Mancozeb is a contact fungicide widely used against numerous plant pathogens in various horticultural crops (e.g., potato, tomato, grapes and citrus) because of its broad-spectrum inhibition activity [5], and its market size is expected to increase further.

Mancozeb is a health hazard for animals/humans. It has many toxic effects such as impaired male sterility in rabbits [6], endocrine-disrupting activities that induce developmental and reproductive effects [7,8], DNA damage [9], and a reduction in sperm count and motility/viability in rats [10] was reported. The water solubility of mancozeb makes it an effective fungicide, but the factors such as light, temperature, humidity, pH influence its stability [11]. It has a significant fatal effect on soil microbiota, nitrification and soil proteases/enzymes due to leaching through soil [12].

Guar gum is a galactomannan polysaccharide produced from guar beans (*Cyamopsis tetragonoloba*) that is a thickening and stabilizing agent in food, feed and industry. Due to intermolecular hydrogen bonding, guar molecules aggregate in aqueous media [13]. Bio-polymers can be used to increase stability, avoid photodegradation and targeted delivery of active ingredients of pesticides due to the smaller size of GG nanoemulsion, which can easily penetrate the fungal cell wall, viscous nature and delayed release, which enhances the absorption time for plants, thereby minimizing soil or water pollution [14,15].

As an emerging field in agriculture sciences, nanotechnology is vital in precision and sustainable agriculture. This encapsulation can protect the active component from environmental factors such as heat, pH, light, moisture, oxygen and hydrolysis [16]. These nanoemulsions/nanoparticles are used in a variety of roles such as targeted delivery [15], slow release of active ingredients [17], lowering toxicity [15], managing fungal diseases [18,19] and weed management [20] in plants. Synthesis of nanoparticles using biological entities such as plants or fungi was reported and is considered the best method of nanoparticle synthesis because the proteins and enzymes present in these living entities act as chelating and capping agents to give small and uniform-sized nanoforms with good antifungal and antibacterial efficacy [21–23].

According to the existing literature, there is an information gap associated with the guar gum nanoemulsion, its efficacy against fungal diseases in plants and its effects on plant growth parameters. As a result, the current investigation was carried out to synthesize mancozeb encapsulated guar gum nanoemulsion (GG) and assess their efficiency against phytopathogens, both in vitro and in vivo. Fungicide-loaded NEs were tested for sustained release behavior, and toxicity was evaluated on Vero cell lines for its non-target harmfulness.

## 2. Materials and Methods

### 2.1. Reagents, Plant Materials and Fungal Cultures

Guar gum, glutaraldehyde (25%), glycerol, dichloromethane, span-80, tween-20 were bought from Hi-Media Pvt. Ltd., Mumbai, India. A commercial fungicide (mancozeb) was purchased from the local dealer. The Vero cell line (derived from African green monkey kidney cells) used to mimic normal mammalian cells was preserved at National Research Center on Equines, Hisar. The Arun Hisar and Kufri Pukhraj variety of tomato and potato seeds, respectively, were supplied by the CCSHAU, Hisar vegetable division. Pathogenic fungi (ITCC3640, ITCC5431, ITCC5492, ITCC6343) were purchased from the Indian Type Culture Collection (ITCC), Division of Plant Pathology, IARI, New Delhi and were revitalized by purification.

### 2.2. Synthesis of Blank and Mancozeb-Loaded Guar Gum Nanoemulsion

Nanoemulsions were synthesized following a previous researcher [23] via an oil-in-water (*o/w*) emulsification method with some modifications. Dichloromethane, 10 mL each, were taken in three conical flasks. A certain amount of mancozeb was added to each flask, followed by 5.0 mL of span-80 under constant stirring to make an oil phase. To the oil phase, 40 mL of 0.5% aqueous guar gum solution was added under continuous stirring to make the final concentration of mancozeb to 0.5, 1.0 and 1.5 mg/mL in respective flasks.

The mixture was rapidly stirred for 5 min. Glycerol (25 mL) was added as a stabilizer, followed by 0.25 mL glutaraldehyde (25%) to effect cross-linking. Nanosuspension was kept overnight at room temperature to form nanoemulsion. Next morning, centrifuged the suspension at 20,000 rpm at 4 °C for 30 min, washed with 15 mL distilled water and recentrifuged. The nanoemulsion (precipitate) was lyophilized, harvested in Eppendorf tubes and preserved in a vacuum desiccator for further use.

The same procedure synthesized Blank NE except for adding mancozeb in the above process.

### 2.3. Depiction of Nanoemulsion via Physio-Chemical Techniques

In the next stage, physio-chemical techniques were used for particle analysis for various parameters.

### 2.3.1. Particle Size, PDI and Zeta Potential

The synthesized NEs were characterized using Zetasizer Nano ZS90 (Malvern Instrumentations, Worcestershire, UK). The sample was disseminated in 950 µL double distilled $H_2O$, and the percentage amount was obtained for 1.0 mL at 25 °C following a previous procedure [24]. The size of mancozeb-loaded guar gum NEs was optimized using a central composite design using Design-Expert Software (Version 13, Stat-Ease Inc., Minneapolis, MN, USA).

### 2.3.2. Fourier Transform Infrared Spectroscopy (FTIR)

Fourier transform infrared (FTIR) spectroscopy was accomplished to find the ionic interactions between fungicide and polymeric nanoparticles with a 5.0 mg fine particles sample using an AVATAR 370 (Thermo Nicolet, San Jose, CA, USA) spectrometer with KBr in a 1:10 ratio amid 4000 and 400 cm$^{-1}$ with a resolution of 4 cm$^{-1}$ and analyzed using online software Spectroscopic Tools, 2019.

### 2.3.3. Transmission Electron Microscope (TEM)

Sonicated samples (set by accumulating a droplet of liquid nanoemulsion on a C-layered Cu-grid trailed by aeration) were subjected to an operating voltage of 200 kV with the Tecnai™ TEM (Thermo Fisher Scientific, Waltham, MA, USA). TEM helps in seeing the internal structure and size of tiny particles.

### 2.3.4. X-ray Diffraction Spectroscopy (XRD)

The sample powder was XRD measured with a D8 Advance Diffractometer (Bruker AXS, Karlsruhe, Germany) in step scan mode with a tube voltage of 40 kV at 40 mA. The samples were scanned in the 5 to 40° area to find the geometric arrangement of atoms (crystalline or amorphous nature of samples).

### 2.3.5. Thermal Analysis Using Differential Scanning Calorimetry (DSC) and Thermogravimetric Analysis (TGA)

Differential scanning calorimetry (DSC, 4000 System, Perkin Elmer, Waltham, MA, USA) generated thermographs from a 3.0 mg sample at LPU, Jalandhar CIF facility. The instrument used an alumina-coated aluminum furnace with a 10 °C/min heating and cooling rate for heating applications ranging from 30 to 445 °C. Unadulterated nitrogen gas (99.99% pure) was poured into the system at a 20 mL/min flow rate.

CIF at Lovely Professional University, Jalandhar, employed a Thermo Gravimetric Analyzer (TGA 4000, Perkin Elmer, Billerica, MA, USA) for TGA. The instrument was heated at a amount of 10 °C/min from 30 to 445 °C. Clean nitrogen gas maintained the system's inertness at a flow rate of 20 mL/min.

### 2.3.6. Encapsulation Efficiency (%) and Loading Capacity (%)

Polymeric nanoemulsions (blank and mancozeb-loaded) were centrifugated at 15,000 rpm for 35 min to evaluate encapsulation efficiency and loading capacity using for-

mula and methods as determined by the same researchers using chitosan-carrageenan NPs in previous research [15] using a UV-Vis spectrophotometer (NanoDrop2000c, Thermo Fisher Scientific, Wilmington, DE, USA).

### 2.4. Evaluation of In Vitro Antifungal Activity

In vitro, antifungal activity was tested on potato dextrose agar (2.0%) via the mycelium-inhibition method. Different concentrations (0.5, 1.0 and 1.5 ppm) of nanoemulsion in an aqueous solution were utilized. A mycelial disc of uniform size (diameter, 5.0 mm) of test pathogens was placed in the middle of test Petri plates and incubated at $28 \pm 1.0\,^{\circ}\text{C}$. The inoculation plates were compared to the control after four days to compute the per cent inhibition rate of pathogen mycelia by means of the equation:

$$\text{Percentage inhibition rate} = \text{Mc} - \text{Mt}/\text{Mc} \times 100 \tag{1}$$

where Mc represents the control mycelial growth and Mt represents the treatment mycelial growth.

### 2.5. Sustained Release Mechanism of Mancozeb

At $37\,^{\circ}\text{C}$, in vitro release behavior of mancozeb from guar gum nanoemulsion was observed for ten h using dialysis tubing (Hi-Media) at pH 7.2 and a standard curve by following the same methodology of previous research by the same authors [15].

### 2.6. In Vivo Antifungal Efficacy of Nanoemulsion in Pot House Conditions

In vivo efficacy of nanoemulsion was tested by the same method and procedure as explained in previous research by the same authors [15]. For bio-efficacy of nanoemulsion, 10 ppm, 15 mL/pot was sprayed on diseased plants and commercial fungicide was used as a positive control at same concentration.

### 2.7. Study the Effects on Plant Growth Parameters

Germination percentage (with three–four true leaves), root–shoot length (at harvesting from root–shoot junction) and dry biomass (without fruit/stolon; oven-dried at $40\,^{\circ}\text{C}$ for seven days) were recorded in triplets.

### 2.8. Cell Viability/Cytotoxicity of Nanoformulation against Vero Cell Line

A Resazurin test was used to assess nanoemulsion cytotoxicity. Cell viability was determined using absorbance because the shift in color from navy blue to pink was directly proportional to cell vitality using the method in an earlier study by the same researchers [24]. Pink-colored resorufin was formed in living cells due to the reductase activity of mitochondria, and optical density was detected by a spectrophotometer (ELISA plate reader) at 590 nm.

### 2.9. Statistical Study

All the data in the findings are provided as the mean $\pm$ standard deviation. A *t*-test was used to calculate the association between data points, with a *p*-value of 0.05 being the least significant.

## 3. Results and Discussion

### 3.1. Physio-Chemical Properties of Synthesized Nanoemulsion

#### 3.1.1. Size Optimization and Stability of Nanoemulsion

Particle size is an essential parameter in nanoparticle synthesis and is influenced by many parameters such as polymer concentration, stirring time and speed. The optimization graph (Figure 1) indicates that when mancozeb concentrations increased, particle size first increased and then decreased. In the instance of guar gum, the impact was less noticeable. The particle size was largest at intermediate concentrations of mancozeb and most minor

at intermediate concentrations of guar gum. RSM optimization values are denoted in Supplementary Table S1.

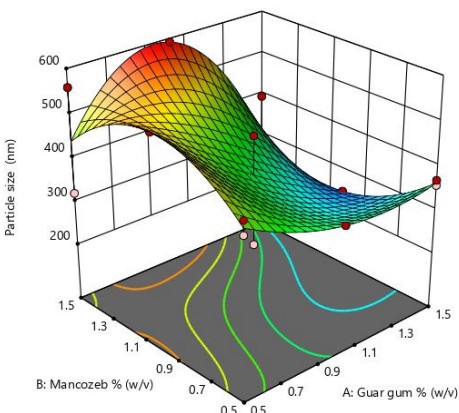

**Figure 1.** Response surface methodology (RSM) plots for size optimization of nanoemulsion for guar gum and mancozeb dose.

The average size of blank NEs, i.e., those without mancozeb loading, was 186.5 ± 1.8 nm, whereas NEs with 1.0 mg/mL mancozeb (GG-1.0) had a larger size (246.6 ± 0.9 nm), as shown in (Figure 2 and Table 1).

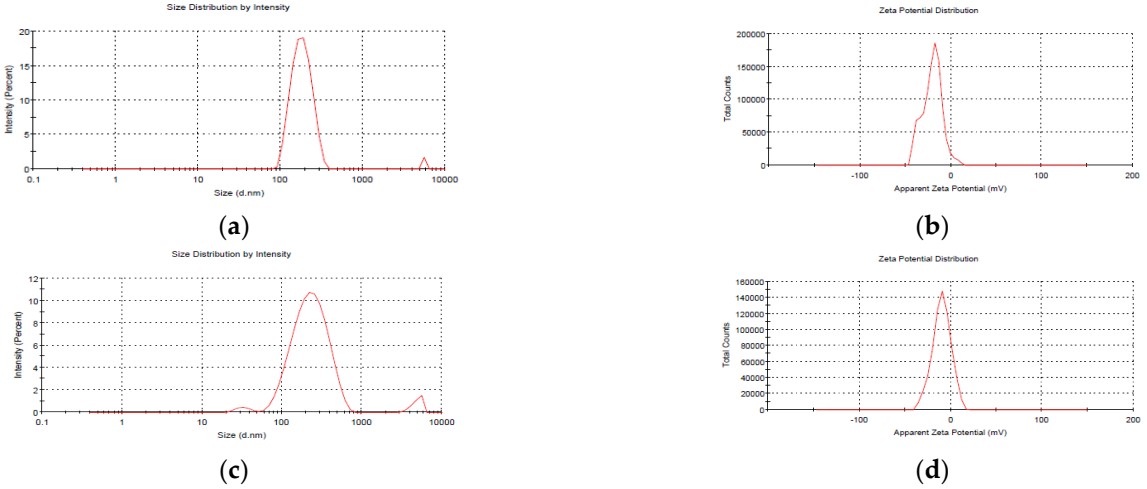

**Figure 2.** Particle size analysis (PSA) of freshly prepared blank GG nanoemulsions (**a**) size (**b**) zeta potential; and mancozeb (1.0 mg/mL) loaded nanoemulsions (**c**) size (**d**) zeta potential.

**Table 1.** Size, zeta potential and polydispersity index (PDI) of GG NEs.

| | Freshly Prepared Nanoemulsion (NE) | | |
|---|---|---|---|
| **Nanoemulsion** | **Size (nm)** | **Zeta Potential (mV)** | **PDI** |
| Blank guar-gum NEs | 186.5 ± 1.8 | −20.4 ± 0.5 | 0.435 ± 0.1 |
| Mancozeb (1.0 mg/mL) loaded guar-gum NEs | 246.6 ± 0.9 | −9.80 ± 0.4 | 0.323 ± 0.2 |
| | Twenty days storage stability of GG NEs in distilled water at 4 °C | | |
| Blank guar-gum NEs | 216.7 ± 2.1 | −9.14 ± 0.2 | 0.360 ± 0.1 |
| Mancozeb (1.0 mg/mL) loaded guar-gum NEs | 394.1 ± 1.1 | −9.42 ± 0.9 | 0.221 ± 0.2 |

Mean ± standard deviation in replication of three.

A PDI 0.435 for blank nanoemulsion showed the manufacture of monodispersed NEs, and zeta potential values up to 30 mV were a good predictor of nanoparticles (NPs) stability [25,26]. The zeta potential values in the current investigation were in the range of −9.14 ± 0.2 to −20.4 ± 0.5 mV, showing the high stability of stored nanoemulsion (Table 1).

After 20 days of storage, GG-1.0 NEs showed a modest change in size (246.6 ± 0.9 nm to 394.1 ± 1.1 nm), which might be attributed to molecular weight, stirring duration, solvent utilized, pH and homogenization speed of the polymeric nanosystem [27].

### 3.1.2. Fourier Transform Infrared (FTIR) Spectroscopy

A prominent peak in raw guar gum polymer at 3416.41 cm$^{-1}$ indicated wide O-H stretching vibration (associated) of sugar units in the guar gum, whereas peaks at 2924.89 cm$^{-1}$ indicate-C-H sym. stretching vibration and, at 1650.63 cm$^{-1}$, are attributable to C-HO-H bending vibrations [28]. Firm peaks in blank GG NEs at 3461.32 cm$^{-1}$ indicate asymmetrical NH$_2$ stretching vibration, whereas at 1637.49 cm$^{-1}$, primary amides were present (Figure 3). The peak shift from 1637.49 cm$^{-1}$ in blank NEs to 1640.97 cm$^{-1}$ in mancozeb-loaded GG NEs indicated mancozeb loading in the bio-polymeric guar gum. Characteristic broad bands of guar gum were acknowledged in the blank and loaded nanoemulsion spectrum, supporting nanoemulsion formation.

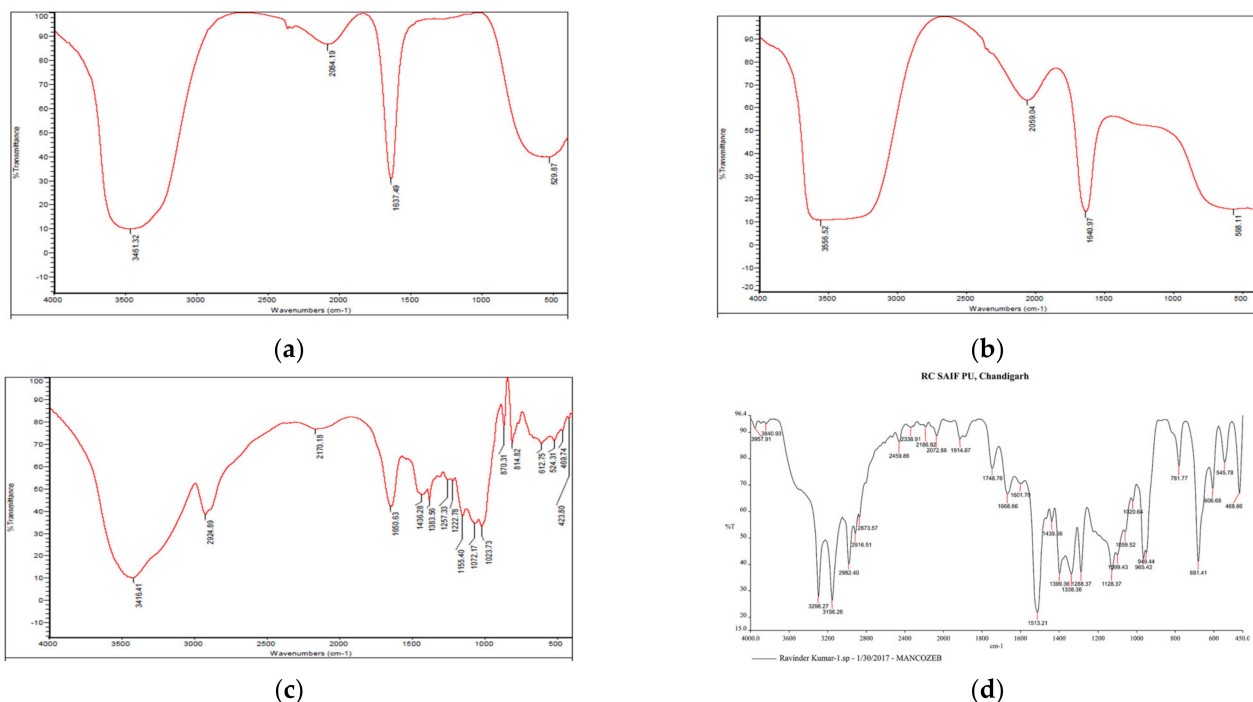

**Figure 3.** FTIR spectra of (**a**) blank GG NEs (**b**) mancozeb-loaded GG NEs (**c**) guar gum (**d**) mancozeb.

### 3.1.3. X-ray Diffraction (XRD) Spectroscopy

Mancozeb exhibited firm peaks, as seen in Figure 4, indicating that it is a highly crystalline substance. On the other hand, raw guar gum polymer displayed a prominent peak at 2θ = 22°, indicating that guar gum is amorphous. Broad guar gum peaks of raw polymer and sharp crystalline mancozeb peaks were buried behind broad guar gum peaks in mancozeb-loaded NE. Maluin et al. [29] also noticed this pattern in the XRD pattern of chitosan-dazomet nanoemulsion. Mancozeb exhibited strong mountains at diffraction positions of 19°, 29°, 38°, and 41°, suggesting crystalline nature.

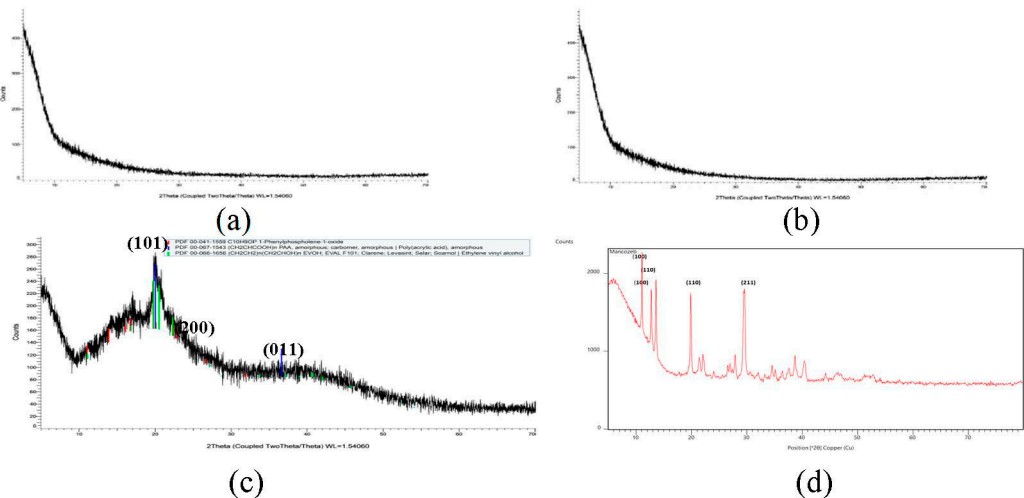

**Figure 4.** XRD spectra (**a**) blank GG nanoemulsion (**b**) mancozeb-loaded GG NEs (**c**) guar gum (**d**) mancozeb.

### 3.1.4. Differential Scanning Calorimetry (DSC)

Figure 5a depicts the DSC curve of natural GG and generated nanoemulsion (Figure 5b,c), revealing that both materials exhibited an endothermic peak below 100 °C due to moisture loss [30]. In the case of GG, an exothermic peak at around 275 °C was seen, indicating that the polysaccharide structure was being degraded, and the succeeding products, such as CO, $CO_2$ and $CH_4$, were more stable as compared to polysaccharides. Breaking (d)-galactose and (d)-mannose units of the guar gum molecule may result in a large peak of guar gum breakdown at 300 °C [31].

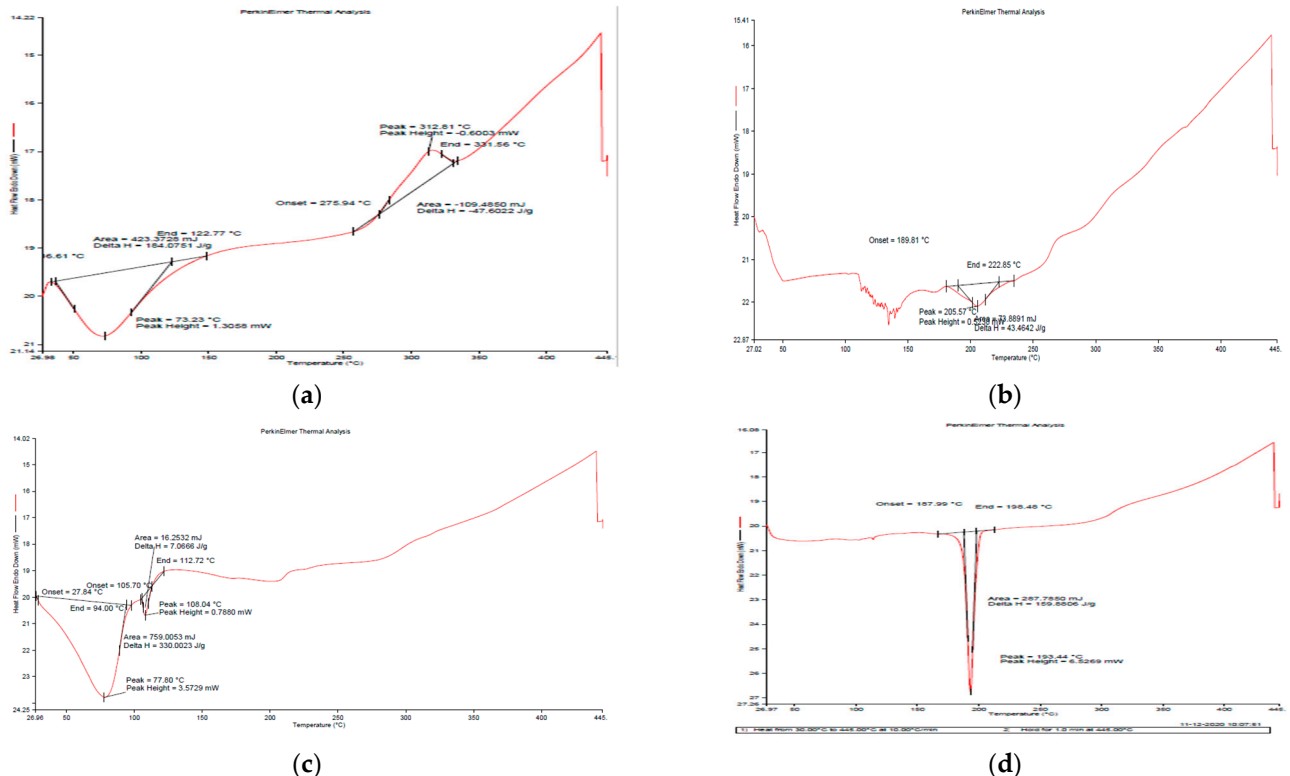

**Figure 5.** DSC thermographs of (**a**) guar gum (**b**) blank GG NEs (**c**) mancozeb-loaded GG NEs (**d**) mancozeb.

### 3.1.5. Thermogravimetric Analysis (TGA)

The degradation of commercial fungicide into dualistic phases along with sulfuric gas emissions around 200 °C is depicted in Figure 6. The first weight loss of 26.5 percent between 25 and 200 °C was related with $CS_2$ and hydrogen sulphide ($H_2S$) emissions with minor $SO_2$ and carbon monoxide levels. The second weight loss (22%) occurred amid 200 °C and 260 °C with $H_2S$ release. The first weight loss (20%) occurred in raw guar gum powder at 25–270 °C owing to $H_2O$ loss, and the second weight loss (60%) occurred from 270 to 350 °C.

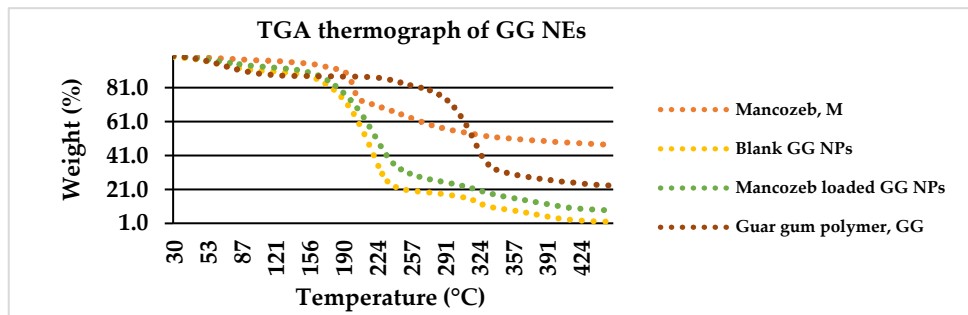

**Figure 6.** Weight loss percentage of mancozeb, blank and loaded nanoemulsion as a function of temperature via TGA thermograph.

In previous research, polymer encapsulation improved the thermal stability of active constituents [32]. For blank GG NEs, the first weight reduction occurred at 160 °C owing to water loss [33], and the second (70 percent loss) occurred from 160 to 240 °C due to GG NE decomposition. For mancozeb-loaded NEs, a similar configuration was found where mancozeb and NEs decomposed concurrently [34].

### 3.1.6. Loading Confirmation of Mancozeb via TEM

TEM graphs revealed circular-shaped NEs (102 nm) with a blank interior for dummy NEs and a dark, dense spherical shape of 101–164 nm for mancozeb loaded, confirming the loading of mancozeb inside the nanoemulsion as shown in Figure 7a,b [15].

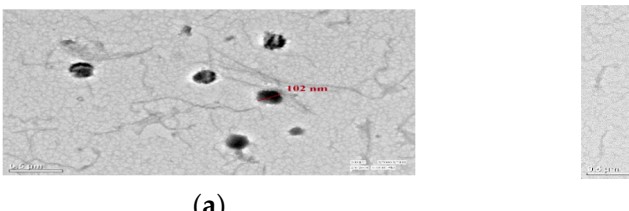    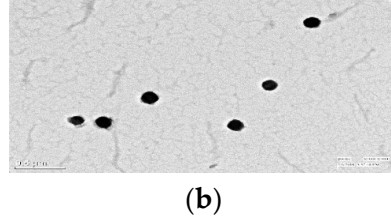

(**a**)                                      (**b**)

**Figure 7.** Transmission electron microscopy of GG nanoemulsion of 102 nm size (**a**) blank (**b**) fungicide-loaded; at ×7000 magnification, 200 kV, 0.5 μm scale.

### 3.2. Encapsulation Efficiency (EE) and Loading Capacity (LC)

The Encapsulation efficiency percentage (EE) ranged from $38.3 \pm 0.71\%$ to $58.3 \pm 0.85\%$, while the LC was from $14.3 \pm 0.94\%$ to $18.2 \pm 0.59\%$ for pure and laden GG NEs, respectively (Table 2). Loading capacity (LC) in the present study was found lower than fungicide-loaded chitosan–carrageenan NEs, which may be due to higher ionic interactions between positively charged $NH_4^+$ groups of chitosan and negatively charged $OH^-$ of carrageenan and, thus, higher EE and LC [35]. H-bonding between guar gum polymer-mancozeb enhanced the loading of mancozeb in nanocarrier [36].

**Table 2.** Mancozeb encapsulation efficiency (EE) and loading capacity (LC) of GG NEs.

| Nanoemulsion | EE (%) | LC (%) |
|---|---|---|
| GG-0.5 | 38.3 ± 0.71 | 14.3 ± 0.94 |
| GG-1.0 | 38.2 ± 1.58 | 16.2 ± 0.34 |
| GG-1.5 | 58.3 ± 0.85 | 18.2 ± 0.59 |

Mean ± standard deviation values in replication of three.

### 3.3. Release Behavior of Encapsulated Mancozeb from Nanoemulsion

Figure 8 depicts that nearly 98.0% of the marketable mancozeb was released in just 2.0 h while only 43.0% was released for all the mancozeb-loaded nanoformulations (0.5, 1.0 and 1.5) at the same time. After 8.0 h, it was 76.0% for GG-1.5, while the remaining nanoformulations showed 63.0% aggregate discharge. Earlier, a sustained release was demonstrated by organosilica [37] nanoparticles for agricultural applications. A neutral pH (7.2) was chosen for the release behavior study because the nanocarrier may be hydrolyzed in basic conditions, and the amide (–CONH–) bonds might be cleaved under acidic conditions [38].

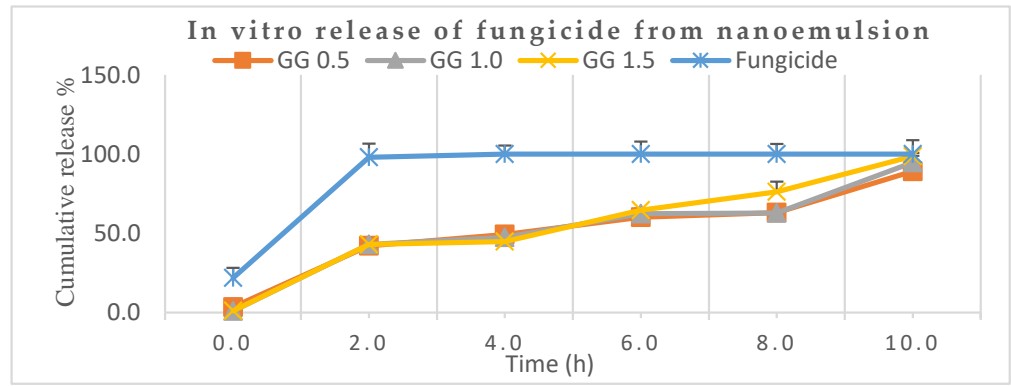

**Figure 8.** In vitro release behaviors of mancozeb from guar gum (GG) nanoemulsion in phosphate-buffered saline, PBS (pH 7.2).

In another trial, thifluzamide-loaded mesoporous silica nanoparticles alleviated the adverse effects of fungicide and promoted purine and pyrimidine metabolism, showing a sustained release mechanism of 40 h [39]. The advantage of guar gum is that it is food-grade quality, viscous and plays a pivotal role as an eco-friendly adjuvant for sustained discharge of fungicide not cause harm to non-target entities.

### 3.4. In Vitro Antifungal Activity

GG-1.0/1.5 and mancozeb (at 1.0 and 1.5 ppm) exhibited complete mycelial inhibition against *S. lycopersici* and *S. sclerotiorum.* An 84.5% inhibition was shown by GG-1.5 which is quite comparable to commercial mancozeb (86.5 ± 0.7%) at the same concentration. The highest mycelial growth inhibition by blank NEs was demonstrated against *A. solani* followed by *S. lycopersici* (Table 3). The mycelium inhibition plates are shown in Supplementary Figure S1. The current study results align with an earlier study where Cu–chitosan nanoparticles repressed mycelial growth by 50–52.7% [40]. Earlier, mancozeb loaded chitosan nanoparticles exhibited good antifungal efficacy against *F. pallidoroseum* [41].

**Table 3.** In vitro inhibition percentage of GG nanoemulsion against fungal phytopathogens.

| Fungi | Nanoformulation with Mancozeb (ppm) | GG NEs Fungus Diameter (mm) | GG NEs % Inhibition = dc − dt/dc × 100 | Mancozeb (ppm) | Mancozeb Fungus Diameter (mm) | Mancozeb % Inhibition = dc − dt/dc × 100 |
|---|---|---|---|---|---|---|
| *A. alternata* (ITCC6343) | Blank NEs, N 1.0 | 29.0 ± 1.4 | 62.6 ± 1.4c | | -- | -- |
| | Loaded NEs, NF 0.5 | 26.0 ± 1.4 | 66.5 ± 1.4c | F 0.5 | 12.0 ± 1.4 | 84.5 ± 1.4b |
| | Loaded NEs, NF 1.0 | 13.0 ± 0.0 | 83.2 ± 0.0b | F 1.0 | 11.5 ± 0.7 | 85.2 ± 0.7b |
| | Loaded NEs, NF1.5 | 12.0 ± 0.0 | 84.5 ± 0.0b | F 1.5 | 10.5 ± 0.7 | 86.5 ± 0.7b |
| *S. lycopersici* (ITCC5431) | Blank NEs, N 1.0 | 13.5 ± 0.7 | 59.1 ± 0.7c | | -- | -- |
| | Loaded NEs, NF 0.5 | 16.5 ± 0.7 | 60.0 ± 0.7c | F 0.5 | 14.5 ± 0.7 | 56.1 ± 0.7c |
| | Loaded NEs, NF 1.0 | 00.0 ± 0.0 | 100 ± 0.0a | F 1.0 | 00.0 ± 0.0 | 100 ± 0.0a |
| | Loaded NEs, NF 1.5 | 00.0 ± 0.0 | 100 ± 0.0a | F 1.5 | 00.0 ± 0.0 | 100 ± 0.0a |
| *A. solani* (ITCC3640) | Blank NEs, N 1.0 | 21.5 ± 0.7 | 66.9 ± 0.7b | | -- | -- |
| | Loaded NEs, NF 0.5 | 21.0 ± 1.4 | 67.7 ± 1.4b | F 0.5 | 10.5 ± 0.7 | 83.8 ± 0.7b |
| | Loaded NEs, NF 1.0 | 19.0 ± 1.4 | 70.8 ± 1.4b | F 1.0 | 10.0 ± 0.0 | 84.6 ± 0.0b |
| | Loaded NEs, NF 1.5 | 10.5 ± 0.7 | 83.8 ± 0.7b | F 1.5 | 10.0 ± 0.0 | 84.6 ± 0.0b |
| *S. sclerotiorum* (ITCC5492) | Blank NEs, N 1.0 | 14.0 ± 0.0 | 58.8 ± 0.0c | | -- | -- |
| | Loaded NEs, NF 0.5 | 14.5 ± 0.7 | 57.4 ± 0.7c | F 0.5 | 10.5 ± 0.7 | 69.1 ± 0.7c |
| | Loaded NEs, NF 1.0 | 00.0 ± 0.0 | 100 ± 0.0a | F 1.0 | 00.0 ± 0.0 | 100 ± 0.0a |
| | Loaded NEs, NF 1.5 | 00.0 ± 0.0 | 100 ± 0.0a | F 1.5 | 00.0 ± 0.0 | 100 ± 0.0a |

Each value is the mean of triplicates. Mean ± SD followed by the same letter in the treatment column are not significantly different at $p \leq 0.05$ as determined by a *t*-test. Abbreviations: N = blank NE; F = fungicide; NF = fungicide loaded NE and 0.5, 1.0 and 1.5 indicates concentration.

*3.5. In Vivo Bioefficacy of Nanoemulsion in Pot House Conditions*

Pathogen-treated fungicide encapsulated nanoemulsion (NFP) revealed a disease control efficacy (DCE%) of 68.5 ± 1.9, 67.7 ± 0.5, 70.4 ± 2.2, 67.5 ± 2.3 against early blight, leaf spot (in tomato) and early blight, stem rot of potato, respectively, which is comparable or higher than the pathogen-treated commercial fungicide, FP [42]. Maximum early blight (08.7 ± 1.6) disease severity (DS%) in potatoes was seen in the case of fungicide-loaded NEs and 14.6 ± 3.4% in pathogen-sick plants treated with a commercial fungicide. It is apparent from Table 4 that NEs displayed an enhanced disease control efficacy for all test fungi. In vitro mycelium inhibition showed higher DCE (up to 100% inhibition) compared to in vivo pot experiment, which may be due to bulk biomass conditions of soil and plants in pot house experiment.

**Table 4.** In pot house conditions, percentage disease severity (DS) and disease control efficacy (DCE) of guar gum nanoformulations.

| Treatment | Tomato Early Blight (*A. alternata*) % DS | % DCE | Leaf Spot (*S. lycopersici*) % DS | % DCE | Potato Early Blight (*A. solani*) % DS | % DCE | Stem Rot (*S. sclerotiorum*) % DS | % DCE |
|---|---|---|---|---|---|---|---|---|
| Pure control, C | 16.1 ± 1.4 | -- | 12.7 ± 1.5 | -- | 10.5 ± 0.7 | -- | 13.5 ± 2.1 | -- |
| Control + Pathogen, CP | 42.9 ± 3.3 | -- | 40.9 ± 6.8 | -- | 29.4 ± 1.6 | -- | 27.4 ± 1.6 | -- |
| Fungicide, F | 10.1 ± 1.9 | 76.5 ± 5.8a | 10.2 ± 1.8 | 75.1 ± 1.8a | 08.0 ± 0.6 | 72.8 ± 1.1a | 08.7 ± 1.0 | 68.2 ± 3.9a |
| Fungicide + Pathogen, FP | 14.6 ± 3.4 | 66.0 ± 3.5a | 12.9 ± 2.3 | 68.5 ± 1.1b | 09.9 ± 0.5 | 66.3 ± 2.2c | 12.9 ± 2.4 | 52.9 ± 3.4c |
| GG Blank NEs, N | 12.9 ± 4.1 | 69.9 ± 5.3b | 12.2 ± 3.0 | 70.2 ± 2.2a | 07.9 ± 1.6 | 73.1 ± 0.0a | 10.0 ± 0.8 | 63.5 ± 4.1b |
| Blank NEs + Pathogen, NP | 14.1 ± 5.7 | 67.1 ± 5.0b | 14.0 ± 3.7 | 65.8 ± 2.4b | 11.2 ± 2.7 | 61.9 ± 1.2c | 10.7 ± 0.8 | 60.9 ± 3.7b |
| Loaded NEs, NF | 11.1 ± 3.6 | 74.1 ± 3.7a | 10.2 ± 0.9 | 75.1 ± 2.4a | 07.2 ± 1.8 | 75.5 ± 1.5a | 8.0 ± 0.6 | 70.8 ± 3.6a |
| Loaded NEs + Pathogen, NFP | 13.5 ± 2.3 | 68.5 ± 1.9b | 13.2 ± 2.5 | 67.7 ± 0.5b | 08.7 ± 1.6 | 70.4 ± 2.2b | 8.9 ± 0.9 | 67.5 ± 2.3a |

Each statistic is the average of three replicates. In the treatment column, mean ± SD followed by the same letter indicates that values are not statistically different at $p \leq 0.05$, as determined by a *t*-test.

### 3.5.1. Consequence of NEs Usage on Seed Germination Percentage (GP)

The germination rate of tomato seedlings administered with nanoemulsion was 80 and 78 percent for blank and filled NEs, respectively, whereas the germination rate of plants treated with fungicide was 60 percent. Hence, the NEs positively affected germination percentage; however, an opposite trend was seen in potato plant's germination for loaded NEs, which had some decreased germination and also reported earlier [43]. Another researcher reported an increase in germination percentage and also other seed parameters on treatment with metallic nanoparticles [44].

### 3.5.2. Influence on the Dry Mass per Plant (DMPP)

Dry biomass of the tomato plants increased exponentially for blank and loaded nanoformulation to 1030 mg and 1320, respectively. In contrast, mancozeb-treated seeds revealed dry biomass of a mere 630 mg, and this type of enhanced biomass was also reported earlier [45]. Similar biomass trends were seen in potato plants but to a reduced extent. In earlier research, it was found that seed treatment with iron pyrite nanoparticles increased the dry mass and yield of spinach [46].

### 3.5.3. Outcome of the Root/Shoot Ratio of the Test Plant

Treatment of commercial mancozeb and Nes tends to increase the root length in both plants compared to control plants (without any treatment) and, thus, are equal or improved in action [47]. No direct co-relation could be deduced in shoot length. Earlier, it was found that 40 nm ZnO and $ZnSO_4$ of nanoparticles increased the root/shoot ratio due to enhanced translocation while 300 nm did not change the root/shoot ratio [48]. It was noticed that the nanoform of fungicide was comparable or more effective than commercial mancozeb for the above-said three criteria. These results established that nanoformulation is less harmful or has growth promontory effects on plants than commercial fungicide mancozeb.

### 3.6. In Vitro Cytotoxicity Assessment of Nanocarrier on Vero Cells

The maximum cell viability of 75.60% was achieved for GG-0.5 NE, trailed by GG-1.0 (74.02%) on a 0.25 mg/mL dose, higher than commercial fungicide (66.22%). The most significant results were seen at a 1.0 mg/mL treatment concentration, where wide cell viability gaps were observed for commercial mancozeb (21.67%) and NEs treatments (63.83–71.88%). These results suggest that 1.0 mg/mL is the optimum dose in terms of cytotoxicity perspective (Figure 9). Positive control, DMSO (known to be lethal), showed cell sustainability of 18.98%. Earlier, itraconazole-loaded polymeric nanoparticles showed good antifungal efficacy without toxicity to A549 cells [49]. Contrary to this, some studies also reported toxic effects of nanoparticles on cells, which may be due to ROS generation, DNA damage [50], while some researchers reported ecological/nontoxic nature of these biopolymeric nanoparticles [51,52].

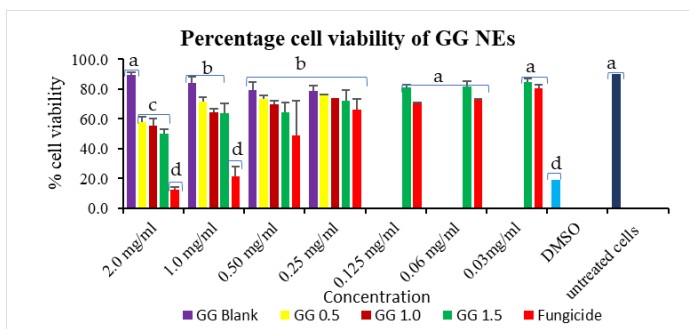

**Figure 9.** GG nanoemulsions percentage cell viability in Vero cells. Column's tops display the standard deviation (SD). A *t*-test indicates that different letters in graph columns have significant levels at $p \leq 0.05$.

Earlier also, while working with Vero cells, nanocarriers were found to be non-toxic to cells for in vitro studies [53,54]. Thus, these in vitro results suggest that nanoparticles are safe and can be used for agricultural applications. However, large-scale field trials and in vivo toxicity studies are mandatory before commercial applications of these nanocarriers in agriculture.

## 4. Conclusions

Chemical fungicides contain many harmful chemical entities which interfere with the non-target species. This study used guar gum to form a nanoemulsion holding mancozeb

via an emulsification method in a size range of 186.5–394.1nm. DSC and TGA recognized the heat strength of nanoemulsion. In vitro antifungal activity was more than or comparable to test fungicide. In the pot house trial, good DCE was seen for both test plants against diseases that may be due to enhanced constituent's translocation because of the nanocarrier's small size. In vitro toxicity assay found good viability of the Vero cell line at 1.0 mg/mL compared to commercial mancozeb. Extended release of mancozeb from the nanocarrier may be the reason for enhanced cell viability. Hence, guar gum nanocarrier with prolonged release and higher bio-safety may become a promising substitute for harmful chemical fungicides.

**Supplementary Materials:** The following supporting information can be downloaded at: https://www.mdpi.com/article/10.3390/jox13020020/s1, Table S1: Runs to optimize the size of guar gum NEs using response surface methodology; Figure S1: In vitro antifungal ability of blank (N3), and fungicide-loaded (N3F) GG NEs against (a) *A. alternata* (b) *S. lycopersici (c) A. solani (d) S. sclerotiorum*; where: C-control, PC-pure control (media only), N3-blank NEs, N3F-mancozeb (1.0 mg/mL) encapsulated NEs and F- commercial fungicide at 0.5, 1.0 and 1.5 ppm. Figure S3: Disease controlling efficacy of GG NEs on tomato plants (a) early blight (b) leaf spot; on potato plants (c) early blight (d) stem rot; in pot conditions; where C–control (No treatment), CP1-P1, F-fungicide, FP1—fungicide+P1, N3—blank NEs, N3P1—blank NEs+P1, N3F—fungicide-loaded NEs, N3FP1—fungicide loaded NEs+P1, CP2—pathogen P2, FP2-fungicide+P2, N3P2—blank NEs+P2, N3FP2- fungicide-loaded NEs+P2.

**Author Contributions:** Conceptualization, R.K., J.S.D. and A.M.; methodology, R.K. and A.M.; software, R.K., M.N. and B.S.S.; validation, J.S.D.; formal analysis, D.K. and P.K.S.; investigation, R.K.; resources, J.S.D. and R.K.; data curation, P.C., P.K.S. and R.K.; writing—original draft preparation, R.K.; writing—review and editing, J.S.D., M.N., D.K., A.M. and B.S.S.; visualization, D.K. and P.K.S.; supervision, J.S.D. and A.M.; project administration, J.S.D. All authors have read and agreed to the published version of the manuscript.

**Funding:** There was no external funding support for this study.

**Institutional Review Board Statement:** Not applicable.

**Informed Consent Statement:** Not applicable.

**Data Availability Statement:** The data presented in this study are available on request from the corresponding author.

**Acknowledgments:** For infrastructural support and bio-physical characterization of materials, the authors thank the Chairperson of the Department of Biotechnology at CDLU Sirsa and the Directors of SAIF at AIIMS New Delhi, STIC Kochi, the Department of Pharmaceutical Sciences at GJUS&T Hisar, and LPU Jalandhar, respectively.

**Conflicts of Interest:** The authors declare no conflict of interest.

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
