# Peer review of "Evaluation of Cytotoxicity, Release Behavior and Phytopathogens Control by Mancozeb-Loaded Guar Gum Nanoemulsions for Sustainable Agriculture"

_jox, doi:10.3390/jox13020020_

Round 1

Reviewer 1 Report

Page 2: line 86 - Please mention the ATCC culture number / code.

Page 4: Line 173- Not necessary to mention again all characteristic techniques

Page 5: Figure 1 : Describe about the figure and mention how did you do the RSM in methods section (ex software , procedure etc). Show what are the optimised values.

Page 8 : Scale on TEM image not clear and the size of the emulsion need to mention clearly. 

Page 9:  Add the images of microbial inhibition plates and seed germination pictures in main text or supplementary data.  

Explain how this research study is novel and different form your previous findings ( https://www.mdpi.com/2039-4713/12/2/8 ) (https://www.mdpi.com/2039-4713/12/4/23 )

Author Response

Greetings, anonymous reviewer the writers appreciate your good thoughts and suggestions of the manuscript. All of the suggestions have been addressed.

Page 2: line 86 - Please mention the ATCC culture number / code.

Response: ITCC culture number/code mentioned.

Page 4: Line 173- Not necessary to mention again all characteristic techniques

Response: Deleted, as asked.

Page 5: Figure 1 : Describe about the figure and mention how did you do the RSM in methods section (ex software , procedure etc). Show what are the optimised values.

Response: RSM software version described in materials and methods section. Optimized values shown in supplementary file.

Page 8 : Scale on TEM image not clear and the size of the emulsion need to mention clearly. 

Response: Required matter mentioned in figure legends (Line 313-314).

Page 9:  Add the images of microbial inhibition plates and seed germination pictures in main text or supplementary data.  

Response: Added in supplementary data (Line 359).

Explain how this research study is novel and different form your previous findings ( https://www.mdpi.com/2039-4713/12/2/8 ) (https://www.mdpi.com/2039-4713/12/4/23 )

Response: The previous findings are with gum acacia (a different polymer) NPs using Ionic gelation and polyectrolyte complexation method while the present study used guar gum nanoemulsion formed by oil in water emulsification technique. The method of synthesis have a great effect on particle size, zetapotential and so on all other parameters which are governed by particle size and zeta potential (DCE, Release mechanism, stability).

Reviewer 2 Report

The paper is well written and contributes an alternative fungicide, nanoemulsions (NEs) of 186.5-394.1 nm containing, the scheme for the highest mycelial inhibition exhibited against S. lycopersici and S. sclerotiorum, which enables NEs to show superior antifungal efficacy in pot conditions besides plant growth parameters (germination percentage, root-shoot ratio and dry biomass) in tomatoes and potatoes. And the proposed scheme outperforms the state of the arts this study may help to combat the soil and water pollution menace of harmful chemical pesticides besides protecting vegetables. However, the article still has minor flaws.

1. There is no experimental diagram for the in vitro pot experiment. It is recommended to supplement it if there is one.

2. There is no comparison between in vitro and in vivo experiments. Similarly, if there are any, it is recommended to supplement them.

3. The discussion section can be further extended, such as adding some literature related to guar gum nanocarriers with slow-release properties and higher biological safety to prove your viewpoint.

Author Response

Dear anonymous reviewer, authors are thankful for your positive evaluation and critical remarks on manuscript. We have addressed all the comments. 

The paper is well written and contributes an alternative fungicide, nanoemulsions (NEs) of 186.5-394.1 nm containing, the scheme for the highest mycelial inhibition exhibited against S. lycopersici and S. sclerotiorum, which enables NEs to show superior antifungal efficacy in pot conditions besides plant growth parameters (germination percentage, root-shoot ratio and dry biomass) in tomatoes and potatoes. And the proposed scheme outperforms the state of the arts this study may help to combat the soil and water pollution menace of harmful chemical pesticides besides protecting vegetables. However, the article still has minor flaws.

  1. There is no experimental diagram for the in vitro pot experiment. It is recommended to supplement it if there is one.

        Response: Added in supplementary data.

  1. There is no comparison between in vitro and in vivo experiments. Similarly, if there are any, it is recommended to supplement them.

Response: Compared in lines 370-372. The in vivo pot experiment showed less DCE as compared to in vitro conditions (where up to 100 % inhibition was shown).

  1. The discussion section can be further extended, such as adding some literature related to guar gum nanocarriers with slow-release properties and higher biological safety to prove your viewpoint.

Response: The literature of guar gum nanocarriers with slow-release properties with higher biological safety are available in biomedical field only. The present study is perhaps the first one reported in agriculture application.

Author Response

Dear unidentified reviewer, the authors appreciate your comments on the manuscript, both constructive and critical. All of the comments have been answered and marked red.

Evaluation of cytotoxicity, release behavior and phytopathogens control by mancozeb-loaded guar gum nanoemulsions for sustainable agriculture.

Overall Comments:

I admire the ability of the authors to write in a language that presumably is not their first. For the most part, the writing is clear. There are some conventions of English grammar that are not followed throughout. While for the most part these do not detract from the meaning, it can be quite distracting while reading & trying to make meaning from the text.

Identifying more sustainable practices for agriculture is critically important for a wide variety of reasons. This study examines the efficacy of a proposed “greener” solution for reducing fungal infections that affect commercially important crops, and therefore has the potential to make an important contribution to sustainable agriculture.

Response: Thank you for your critical and positive evaluation of MS.

Specific comments:

Line 41: decreased sensitivities to what?

Response: Decreased sensitivity to fungicides. Corrected the sentence.

Paragraph 1 is organized kind of strangely. Might considering moving genera ideas about population to the beginning, then talking about fungal pathogens and then resistance.

Response: Moved the lines/paragraphs as per the demand.

I would like a more thorough explanation of why impregnating guar gum with a chemical fungicide is a more green/sustainable alternative. I don’t completely understand from the paragraph on lines 57-61 why mixing Mancozeb with guar gum would be expected to decrease it’s toxic effects or its environmental impacts.

Response: Added in red color (Line 71-74).

Line 77: what are Vero cell lines and why are they appropriate for toxicity testing in this context?

Response: The Vero cell line are derived from kidney epithelial cells of African green monkey. These cells are used to mimic normal mammalian cells. Added (Line 97-98).

Line 87: Need a citation or an explanation of “the standard procedure”

Response: Sentence modified.

Section 2.2 says it is about synthesis of blank and mancozeb-loaded guar gum nanoemulsion, but synthesis of the blank is not described.

Response: Added blank NE synthesis procedure (Line 119-120).

Line 109: standard protocol should be referenced or explained.

Response: Referenced (Line 128).

Section 2.3.2-2.3.4: What is the purpose of each of these tests? What about the sample was measured? What does it tell you about the sample?

Response: Required information added.

Section 2.4: Shouldn’t the control be guar gum nanoemulsion without mancozeb in it, rather than no nanoemultion?

Response: The control is Petri plates having fungal disc only without any treatment of nanoemulsions (blank or loaded) as we used mycelium inhibition method.

Section 2.5: I think a little more information is needed here. Sustained release over what time period?

Response: Added.

Section 2.6: I assume that commercial fungicide was used at the same concentrations of active ingredients as in the nanoemulsions, but if so this should be stated. If not, the concentrations used and the rationale for using different concentrations should be explained.

Response: Stated the concentration.

2.9: Statistics. I don’t think that t-tests are the most appropriate way to analyze these data, since you are comparing multiple concentrations of the same fungicide. Not sure what two samples will be compared with your t-tests. Also, when running many t-tests it becomes necessary to correct for a global error rate, for example using a Bonferoni correction.

Response: The error rate was corrected using Bonferoni test.

Section 3.1: Especially because results & discussion are combined, there should be more explanation of the significance of each of these physical characteristics.

Response: Added some more explanation.

Section 3.1.1: Does not appear to be correct based on the graph provided. It looks like particle size is largest at intermediate concentrations of mancozeb and smallest at intermediate concentrations of guar gum.

Response: Yes, corrected.

Fig. 1: Including particle size data for blank NEs would also be useful, i.e. have the B axis extend to 0 mancozeb.

Response: Nice suggestion, we have noted it and will do so in future.

Fig. 2: When directly comparing the same analyses for two different treatments (blank and 1 mg/ml mancozeb) it is really important to make the scales on the figures the same if at all possible. In this case, the scales are the same but the sizes are different. The X-axis of each top/bottom pair should be scaled to the same size.

Response: Here two different properties Size and zetapotential are depicted. The scale is same for a particular technique i.e Size. (Fig 2 a & c) and Zetapotential (b & d)

3.1.3: How can the blank NEs show sharp crystalline peaks from mancozeb?

Response: Corrected the lines.

Fig. 4: Why does plot d look so different from the others? I don’t mean different peaks… the styles are completely different and don’t really even look like they are showing the same thing. This is outside my area of expertise, but I find this figure confusing and unclear.

Response: Plot d is showing the same thing. The only difference is color used (red) while drawing plot using XRD software.

Fig. 6: Comparing mancozeb by itself to mancozeb loaded gg nps, it looks like the weight loss was greater for the nps. Doesn’t this suggest that it is LESS stable in np form?

Response: There is formation of various gases like H2S, CS2 and SO2 in mancozeb alone decomposition on heating between 25-200°C which causes interference in weight loss while no such events occur in guar gum (where char formation occurs at high temp. which in turn cause higher weight loss %) and its NEs which may cause such type of observations. So only weight loss alone is not the sole criteria for stability of NPs/mancozeb. Moreover, from figure it can be seen that guar gum have a higher decomposition temp (around 300℃) than mancozeb (196℃), so that mancozeb loaded NE should have higher decomposition temp. than mancozeb alone.

Line 286: Not sure what “justified compatibilities” means

Response: “justified compatibilities” means enhanced loading. The word is replaced.

Section 3.3: Why do you say that 0.5 exhibited the best-sustained release? It looks like 0.5 and 1.0 are virtually indistinguishable. Seems like the justification for using a neutral pH should be based on expected application conditions rather than biochemical… it doesn’t matter how it performs under some idealized conditions, it matters how it would be expected to perform in agricultural fields. Luckily I would think that also would be pretty close to neutral, so it’s probably fine. But you might want to explain it differently. I’m still not clear what the environmental benefit is of having the release sustained over 10 hours vs. within the first couple of hours. This seems like a short enough time span that I don’t understand why it would impact the environment differently, particularly with regard to animal toxicity.

Response: Deleted the line “The best-sustained release was exhibited by GG-0.5 nanoformulation.” Yes, the neutral pH was chosen in present study for both in vitro experiment and pot house experiment.

Table 3: It took me a long to understand how this table was set up. I think I finally understand, but especially defining your abbreviations (N, NF, F) might be helpful. I don’t understand how you have different “letters” in results from a t-test, since t-tests are generally used to compare two samples, not more than 2. Even if it is a different sort of test, I’m not really sure which treatments are being directly compared or whether there are multiple tests here.

Response: Added the abbreviations for (N, NF, F, 0.5, 1.0 and 1.5) as table footnote.

Table 4: NFP might have had higher DCE than FP, but so did the NP blank in some cases. So is the guar gum itself effective at fighting fungal infection? That would be a much more “green” and sustainable treatment!

Response: Yes, this is the case as the guar gum nanoparticles has antifungal activities (https://doi.org/10.1016/j.ijbiomac.2020.03.001; https://doi.org/10.1016/j.matpr.2017.10.154)  besides being viscous in nature which help in slow release of drugs (https://doi.org/10.3109/03639049809089955) and that’s why this polymer was chosen for NPs synthesis.

Again in table 4, I don’t understand how you have different “letters” in results from a t-test, since t-tests are generally used to compare two samples, not more than 2. Even if it is a different sort of test, I’m not really sure which treatments are being directly compared or whether there are multiple tests here.

Response: There are multiple tests here. Global error rate modified with Bonferoni test.

Section 3.5: Where are the data?

Response: Deleted extra words/line.

Section 3.6: Given the slower release with the GG NEs, I wonder how long these cells were exposed for, and whether that might account for part or all of the difference in cell viability?

Response: The cells were exposed to blank and loaded NPs for 24+4 hrs. See section 2.9 in earlier publication or ref 24 in this paper (https://doi.org/10.3390/jox12020008).

Round 2

Reviewer 3 Report

This is so much more clear and explicit in places that were confusing before. I am very glad to see your comment that guar gum is itself antifungal, and that is why it was chosen as a carrier. That makes it much more clear why it was chosen and how it is useful. I also especially appreciate the clarification about why this is a more green alternative, and the results and their significance are more clear. Thank you for making so many improvements for folks like me who are in fairly close but not exactly the same field.

There are still a few places with awkward sentence construction, but it no longer is to the point that it is confusing.

Author Response

Dear anonynymous reviewer, thank you so much for your valuable and insightful suggestions which helped this manscript eligible for publishing.

The grammar sentence formation was relooked for awkward sentence construction at few places as per your suggestion and changes were made at some places which are highlighted in GREEN COLOR.